# The Relationship between students' engagement, grit, and teacher care in Chinese context

Xin Wang[1], Jianguo Xiao[2]*

**1** Department of Student Affairs, Jilin Normal University, Siping, China, **2** School of Marxism, Jiangsu University, Zhenjiang, China

* xiaojianguo_1981@163.com

## Abstract

The significant growth in the field of English as a Foreign Language (EFL) education, along with the introduction of Positive Psychology (PP), has led to an increased emphasis on student engagement within the academic realm. Additionally, the focus on grit in language education highlights its importance in students' success throughout their educational journey. Besides, learners' engagement in academic domains is a noteworthy supporting the achievement of students in higher education. Also, the investigation of favorable communication actions has been examined as an important indicator of EFL learners' engagement and grit. However, the significance of teacher caring in predicting EFL learners' grit and engagement has been underresearched among Chinese EFL learners and this study makes an effort to consider it. To fill this gap, 401 EFL learners from 11 regions of China were requested to respond to three scales, namely engagement, grit, and teacher care. The data were collected and through Structural Equation Modeling (SEM), it was revealed that about 59 percent of changes in the EFL learners' grit can be predicted by their teacher care, and about 67 percent of changes in the EFL students' engagement can be predicted by their teacher care. Finally, the suggestions for the results are also presented.

## 1. Introduction

While various elements are essential for effective language instruction, psycho-emotional traits appear to have a greater impact on learning and were largely disregarded in constructivism [1]. As a recent development in the field of psychology, known as Positive Psychology (PP), significant focus has been placed on the strength and impact of constructive feelings in achieving a more joyful existence and a successful profession [2–3]. As per the principles of PP, grit, and engagement have a substantial influence on L2 education as they strive to improve educational achievements and cultivate conducive learning environments [4–6]. Learner success was significantly impacted by grit, as a psychological characteristic [3,7,8]. Grit, as a relatively novel concept in the PP framework, includes the theories of passion and

**Data availability statement:** All relevant data are within the paper and its Supporting Information files.

**Funding:** The research is supported by: the General Program Supported by The National Social Science Fund of China in 2020, Research on Improving the Long-term Mechanism of Training Socialist Builders and Successors (No.20BKS140). The funder had no role in study design, data collection and analysis, decision to publish, or preparation of the manuscript.

**Competing interests:** The authors have declared that no competing interests exist.

perseverance, which have received interest in diverse domains, including professional and educational settings [9]. Also, it is said to be a substantial factor in the language instruction procedure and it is recognized in the collection of works as a valuable subject for individuals who are prone to flourish in education as it has been intertwined with enhanced dedication [10–11]. Now, educators and professionals are expected to motivate and cultivate students' abilities and persistence to work vigorously to accomplish their ultimate objectives [12]. Furthermore, Pawlak et al. [13] proposed that gritty individuals are motivated by a personal fascination in the route of acquiring knowledge, and those with individual objectives experience a heightened level of motivation, confidence, and enthusiasm, leading to enhanced achievement, grit, and well-being. While grit has been comprehensively inspected in educational psychology, it has recently been taken into the scrutiny of EFL researchers [13–14].

Further, as PP gained attention in the field of educational psychology, alongside grit, educators comprehended the importance of giving greater attention to the engagement of learners as a crucial element in fostering academic achievement [15–16]. The engagement of students has been extensively studied in various fields [17]. Since learners' engagement has a pivotal part in enhancing their learning success, motivating the learners to actively participate in the learning route has always been a primacy for educators in every educational domain and it is deemed essential for academic achievement, persistence, retention, and academic success [16].

Given that engagement in the pursuit of educational objectives in a language learning setting is of utmost prominence for the academic achievement of EFL learners, it is considered essential to elucidate the favorable factors that foster learner engagement within the realm of EFL investigation [18]. The effectiveness of language acquisition in the educational setting is contingent upon the instructor, and it is asserted that the learner's progress in learning may hinge on the competence of the educators [15]. In self-determination theory (SDT), learners' engagement might be greatly improved through their connection with their instructor [19]. Given the regular exchanges between learners and instructors in language classes, instructors are seen as a crucial means of assistance for learners during classroom activities and learning procedures [20]. In the same vein, Duckworth [21] suggested that educators can be considered as the significant factors for the enhancement and advancement of grit. Also, learners' engagement can be improved through a thoughtful combination with other favorable concepts such as encouragement, connection, social assistance, and care. Teachers can assume diverse functions like being mindful of feedback to learners, offering questions to engage them, and reevaluating classroom administration to regulate relations [22].

More specifically, teacher care refers to teachers' spoken and non-spoken efforts to fulfill the emotional and mental requirements of students [23]. Likewise, caring has been described as the emotions, actions, and ideas that arise from an educator's wish to inspire, assist, involve, or motivate their students [24]. In the realm of academics, the notion of 'care' has been extensively studied and is developing as a crucial factor in successful education [15]. Teacher care has a substantial role in improving learners' engagement and success and fostering the social and ethical

growth of learners [25]. Educators can create a caring atmosphere in which communications are more meaningful when compared to other tactics [26]. Focusing on the reciprocal aspect of this concept, it is argued that 'caring' is the action that emerges from a mutual caring connection between student and instructor, where knowledge acquisition occurs by modeling, conversation, and endorsement at the societal levels [27]. Teacher care motivates student-associated skills such as engagement, mental health, and accomplishment [28–29].

Although teacher care has been explored in terms of engagement in general instruction [30–32], it remains an area that lacks extensive investigations in the field of language learning. For example, Pishghadam et al. [15] and Yuan [33] conducted a limited number of inquiries that concentrated on the impact of teacher stroking behavior, which is a key aspect of teacher care, on learners' success in the EFL context. Teacher care not only encompasses verbal and non-verbal communication strategies—such as stroking behavior—but also includes emotional support, empathy, and responsiveness to students' needs. These elements are crucial for creating a sense of belonging and safety, which are essential for optimal learning. Based on the literature, when learners perceive their teachers as caring, they are more prone to involve intensely in the path of learning, show more motivation, and cultivate resilience in the case of difficulties [18]. This implies that teacher care could act as a key element that impacts academic success.

However, there is still a lack of understanding regarding the influence of teacher care on both grit and engagement of learners in the context of EFL students, especially in China. When considering the combined significance of teacher-caring behavior as a form of positive interpersonal communication in language education, alongside learner engagement and grit, it becomes necessary to prioritize the investigation of the association between these three constructs. Moreover, there has been limited research on the correlation between student engagement, grit, and teacher-caring behavior within the realm of language education. This study aims to examine the specific impact of teacher care on students' grit and involvement, to inform strategic interventions and methodologies that promote increased grit and improved engagement in EFL learning environments. By addressing the identified gaps, this research provides significant insights for educators and policymakers, which can aid in the development of more supportive and enriching learning environments for EFL learners. Finally, this can lead to an increase in student grit and engagement.

## 2. Literature review

### 2.1 Student engagement

In line with PP theory, engagement alludes to intense interest, deep engagement, or focus in everyday activities [34]. In such circumstances, individuals are fully absorbed and involved in tasks and utilize their passions and abilities to cope with challenges [35]. This occurs within the learning environment or when learners are involved in their academic tasks. Hiver et al. [36] further suggested that the concept of learners' engagement is complex and encompasses various characteristics such as affective, intellectual, and social. In general, social engagement can be assessed by examining learners' rate of participation, presence, concentration, resilience, and seeking help [37]. Dincer et al. [38] asserted that certain actions like completing assignments, actively participating in class, and engaging with instructors through questioning and responding are connected to social engagement.

Dincer et al. [38] also characterized affective engagement as students' emotional responses in classes. This category of engagement indicates the curiosity, zeal, and delight an EFL learner exhibits in the route of instruction [37]. Furthermore, they delineated intellectual engagement as students' inclination to employ complex learning strategies rather than simple tactics. This kind of engagement also refers to the cognitive progression, the level of effort, and the ability of self-monitoring and evaluation methods [39]. It is posited that to enhance student engagement, educators should cultivate an empathetic and sociable behavior towards the individual characteristics of students, foster learner autonomy, and demonstrate enthusiasm for their academic pursuits [16]. To achieve this objective, educators can assume diverse roles, including facilitating meaningful discourse, exercising discretion in delivering feedback to students, actively listening to students, employing probing questions to engage students, and reassessing classroom management to regulate interpersonal dynamics [24].

## 2.2 Grit

The concept of grit has gained substantial focus from researchers in the domain of language acquisition in recent times [40]. This increased curiosity is the result of grit as a crucial element that has been associated with educational accomplishments and outcomes [41–42]. Grit is defined as the ability to withstand hardship while preserving motivation for long-standing goals [43]. It is emphasized that grit is the capacity to tap into challenges and cope with obstacles. Grit includes the perseverance of effort and passion for enduring long-standing objectives [21,44]. The former alludes to the wish and grit to achieve an objective, and associates grit with resilience, strength of mind, and caution, and it describes the degree to which persons can endure obstacles while maintaining their unique fortitude [7,45]. The latter is the passion for enduring determination alludes to a person's ability to maintain motivation for a long period and it refers to the degree to which people consistently highlight achieving their long-lasting objectives [7].

Khajavy et al. [41] argue that having grit can assist learners in navigating and effectively overcoming probable obstacles and setbacks they may encounter while learning a second language. Indeed, learners who demonstrate grit tend to display elevated levels of accomplishment, as indicated by their higher grade point averages and exceptional performance in educational environments [46–47]. There is a limited amount of research that specifically targets the enhancement of grit. The incorporation of instructional interventions aimed at promoting perseverance and resilience has emerged as a crucial component of the curriculum.

Duckworth [21] posited that the internal factors contributing to the development and advancement of grit encompass the individual's well-being, behaviors, aspirations, and expectations, alongside the influence of caregivers, educators, educational experiences, and socio-cultural environments. Educators are widely recognized as influential contributors to the efficacy of educational systems and student academic achievement, positioning them as pivotal stakeholders within the pedagogical framework. Educators play a crucial role in the decision-making process within educational institutions, particularly in matters concerning students. This has prompted the recognition of the importance of assessing and examining the behavioral, psychological, and instructional attributes of teachers [30]. Educators typically serve as the predominant providers of educational instruction within the school setting, facilitating opportunities for both teacher-student and student-student interactions in the classroom environment while demonstrating a genuine concern for these interactions.

## 2.3 Teacher care

SDT suggests that the care from others, such as parents and teachers, is essential for learners' achievement that this care helps satisfy the three basic psychological needs: competence, autonomy, and relatedness [48]. SDT is an instructional theory of motivation that is based on the important value of learner autonomy [20]. The SDT suggested that care of others such as parents and instructors is essential for students' success [20]. Teacher care alludes to those manners that aid in establishing positive connections between the students and instructors and are signs of acknowledging and addressing the emotional requirements of students [23,49,50]. In the educational setting, the care that teachers show toward students is a crucial element of the teacher-learner relationship [51]. Caring is a sensation, a construction, and a social display that considers all individuals and can be explained as an emotion, motivation, and/or action, reflecting concerns about the affections and needs of others [52]. Teacher care alludes to the use of both verbal and non-verbal cues by the teacher, such as making eye contact, smiling, nodding, answering with respect, and giving their full attention. These actions help to establish an association between the educator and their learners [53].

Drawing from rhetorical/relational objective theory [54], teacher care fulfills learners' psychosocial and emotional requirements for instructor motivation and respect and relates to teachers' compassionate function, approachability, and intimacy with students, thereby fostering the attainment of favorable educational results by students [23,30]. Pishghadam et al. [55] defined language teacher care with three aspects of impartial connections with others, appropriate feedback, and affection. When EFL educators fail to offer an equitable level of care to their students, they establish partial connections with certain students while neglecting others [56]. EFL teacher care can be demonstrated in their feedback, which,

correspondingly, relates to the details given to learners concerning their execution of a language-acquiring assignment that refers to the concept of competence in SDT. Concerning theories of language learning, the significance of feedback is examined in the Interaction Hypothesis of Long and the Sociocultural Theory of Vygotsky by emphasizing the interactions between educators and learners and the communal aspect of learning [57]. The final aspect of teacher care is teacher stroke, alluding to any act performed by language educators to demonstrate their acknowledgment of learners' existence and caring for them in the EFL setting [55].

As mentioned by Zhang [16], caring educators can establish a reliable educational atmosphere in which learners are greatly motivated to engage in educational tasks. Concerning this, Dewaele [58] proposed that educators who pay attention to their students' emotions, affections, and enthusiasms can motivate them to participate in classroom interaction, thus influencing their level of engagement. Additionally, Lavy and Naama-Ghanayim [32] further asserted that educators of language can cultivate confidence in their students, thus motivating them to actively engage in communication within the classroom setting. Caring and empathetic connections with teachers are particularly important for EFL learners, as it is one of the ways to establish a secure and positive relationship between educators and learners, which refers to the concept of relatedness in SDT and it can affect the learners' success. Teacher care is a complex idea that entails listening and showing interest in what learners would like to communicate, taking into account their emotional state, building trust, and assisting them in accomplishing their maximum potential [23].

## 2.4 Empirical studies

Lewis et al [31] studied the function of teacher care on learners' presentation. To achieve this objective, a sample of 300 students was chosen and the researchers administered a scale and an assessment test to the participants. Investigating participants' answers, the scholars demonstrated a favorable correlation between teacher care actions and learners' accomplishments. Correspondingly, Lavy and Naama-Ghanayim [32] investigated the impact of teacher care on learners' well-being, self-confidence, and engagement. To accomplish this, 675 students were given four dependable scales. Examining the outcomes, a robust and advantageous connection was revealed between instructor caring, learners' engagement, well-being, and self-esteem. By the same token, Yan [59] has made efforts to find any interaction between teacher care and immediacy and students' awareness of fairness. In this process, the subjects of this research consisted of 1,178 Chinese EFL learners, ranging in different ages and educational backgrounds. To gather the data, these scales were handed out and the findings showed that there was a favorable correlation between these three factors. It was indicated that both teacher immediacy and care predict how learners perceive fairness in the classroom. This suggests that teachers who are perceived as more caring and display proper types of immediacy when necessary are believed to positively influence learners' perception of fairness in the classroom.

Recently, Derakhshan, Doliński, et al. [60] inspected the concept of teacher care, educator-learner connection, and engagement in striving for educational objectives in EFL settings, as observed by 233 Iranian and 208 Polish learners. The outcomes of the study through Structural Equation Modeling (SEM) confirmed that care and rapport predicted the engagement of learners in striving for academic objectives. Qualitative data validated the predictive relation of care and rapport with EFL learners' engagement in striving for educational objectives. Moreover, Shen and Guo [61] explored the relationship between the level of respect and assistance that learners get from their instructors and the level of grit that the students show in the academic domains. Respect and assistance are two crucial facets of grit, even though the subject of grit was not directly discussed. The study included a sample of 613 people who were studying English in China. According to the findings, it was discovered that instructor admiration, instructor assistance, and grit were associated with each other, and the outcomes indicated that respect and assistance from instructors affect the greater degrees of grit exhibited by the learners. Yuan [33] explored the methods by which enhancing learners' familiarity with stroke in Chinese classes could enhance their degrees of grit. To accomplish his goal, data was collected from a group of 316 EFL learners. The students were provided with three distinct scales to complete, and the findings obtained through conducting regression

analysis indicated that there are favorable connections between stroke and its impacts on students' grit and that these connections are associated with each other. From the reviewed literature, while several studies have explored the aspects of teacher care, engagement, and grit individually, there is limited research specifically examining how these factors interrelate in the context of Chinese EFL learners. This gap highlights the need for further investigation into the predictive role of teacher care on both engagement and grit among Chinese EFL learners. Therefore, the following questions and according hypothese were formulated:

Q1.Is there any significant relationship between Chinese EFL students' engagement, grit, and teacher care?

Q2.To what extent does teacher care predict EFL students' engagement and grit?

H01: There is a significant relationship between Chinese EFL students' engagement, grit, and teacher care.

H02: Teacher care significantly predicts student engagement and grit among Chinese EFL learners.

## 3. Method

### 3.1 Participants

To gather the data, 401 individuals from eleven regions of China were enlisted through convenience sampling as the researcher had access to them from November 1, 2023, to March 25, 2024. These eleven regions refer to diverse geographic, economic, and cultural areas within China, selected to provide a comprehensive understanding of EFL learners across different contexts. These regions were chosen to ensure a diverse sample that accurately represents the demographic and educational diversity within China. To make the research findings applicable to a larger population, the research participants included 209 male and 192 female Chinese EFL students studying English at the senior high school level, where they were preparing for university entrance exams. Their ages varied from 18 to 33, with a mean age of about 26. All participants provided informed consent, and all data were obtained through the respondents' voluntary cooperation.

### 3.2 Instruments

**3.2.1 Student engagement scale.** This scale, developed by Appleton et al. [62], comprises 35 statements and utilizes a four-point Likert scale (fluctuating from strongly disagree to strongly agree). It aims to assess three distinct aspects of engagement: emotional, behavioral, and cognitive. Emotional engagement focuses on the bond between educators and learners, and the support received from peers and guardians. This aspect measures the quality of the relationship between students and their teachers and it looks at the encouragement, help, and emotional backing students get from their immediate social circles, which can influence their engagement in school. The other is behavioral engagement that measures participation in academic and extracurricular activities.This component evaluates students' involvement in academic tasks such as attending classes, completing assignments, and participating in discussions. Cognitive engagement assesses the quality of school assignments, goals and desires, and intrinsic motivation. Indeed, it examines students' internal drive to learn and succeed.The reliability of this questionnaire was found to be.879 through Composite Reliability (CR).

**3.2.2 Grit scale.** The scholars utilized this scale as an approximately concise Likert scale eight-item inventory which consists of two sections, specifically steadiness of interest and perseverance of efforts. It was created and validated by Duckworth and Quinn [63] wherein items are graded on a 5-point scale from (never like me to very similar to me) to evaluate the level of grit in an individual. It should be noted that the reliability of the questionnaire by running CR was.826.

**3.2.3 Teacher care scale.** The scale was developed by Pishghadam et al. [55], evaluating the beliefs that learners gain regarding the level of care shown to them by one of their educators. The questionnaire consists of 20 statements, assessing the three concepts of stroke (10 statements), feedback (five statements), and relation (five statements). It is a Likert type with six-point options, ranging from (never to always). The reliability of the questionnaire was.956 through CR.

## 3.3 Procedure

Before the process of data collection, a rigorous translation process done by proficient Chinese language experts was conducted to lessen potential risks of misinterpretation or ambiguity in the survey. This process entailed an initial translation in one direction, followed by a subsequent translation in the opposite direction to verify the accuracy of the scales' content. In essence, a backward translation was done, where the Chinese version was translated back into English by two experts. This approach allowed for a thorough comparison between the two versions, warranting that the meanings of the items were well-maintained.The validity of the instrument was verified with oversight from two experts, thus adding a layer of control to bolster its robustness.

Prior to administering the scales, the research team conducted an informative session for the participating students. During this session, the overall objective of the research and detailed guidelines for completing the scales were explained to the students. The objectives of the study and guidelines were obviously stated, with the scales administered in a controlled classroom atmosphere. Written informed consent was acquired from the participants, guaranteeing the privacy of the information they supplied to the investigators and assuring them that they could withdraw from the study at any time. The three scales were formulated and given to the students, who were instructed to evaluate their instructor in terms of the regularity of their caring actions, and subsequently to assess their level of engagement and grit. Data was securely stored and anonymized to protect participant confidentiality, and statistical methods were used to examine relationships between teacher behavior and student engagement and grit. Following the completion of the scales, the data was meticulously collected. The anonymity of participants was strictly maintained by aggregating responses and removing any identifiable information.

## 3.4 Statistical analysis

The data were analyzed using Structural Equation Modeling (SEM) to examine the relationships between teacher care, student engagement, and grit. Descriptive statistics, including means, standard deviations, and frequency distributions, were first calculated to provide an overview of the participants' responses. The internal consistency of each instrument was confirmed through Composite Reliability (CR). SEM was then employed to test the hypothesized model and assess the direct and indirect effects of teacher care on engagement and grit. Linear Regression analysis was also conducted to further explore the predictor roles of variables.

## 4. Results

Along with the aims of the research, the data were inspected through SEM to find the association among the concepts of the study such as learners' engagement, grit, and teacher care, the details was shown in Fig 1.

The result in Table 1 indicated that the ratio of CMIN-DF is 2.686 (spec. ≤ 3.0), goodness-of-fit index (GFI) is 0.932 (spec. > 0.9), comparative fit index (CFI) = .930 (spec. > 0.9), Parsimonious Normed Fit Index (PNFI) = 0.609 (spec. > 0.5), Tucker–Lewis Index (TLI) = 0.920 (spec. > 0.9), and root mean square error of approximation (RMSEA) = 0.065 (spec. < 0.080). The thresholds in Table 1 are the ones proposed by Hu and Bentler [64], according to which the obtained model fit indices are considered satisfactory to excellent.

The results of Table 2 show that the composite reliabilities of the factors are satisfactory (CR > 0.70). In other words, the model meets the standard value for CR. The values also disclose that the convergent validity of the factors reaches a satisfactory value (AVE > 0.50) or that the model has achieved convergent validity. The results of discriminant validity were assessed and are reported in Table 2. Specifically, discriminant validity is evaluated by comparing the square root of the Average Variance Extracted (AVE) for each construct with the inter-construct correlations (Fornell-Larcker criterion). In Table 2, the diagonal values representing the square root of AVE are higher than the inter-construct correlations, indicating satisfactory discriminant validity.

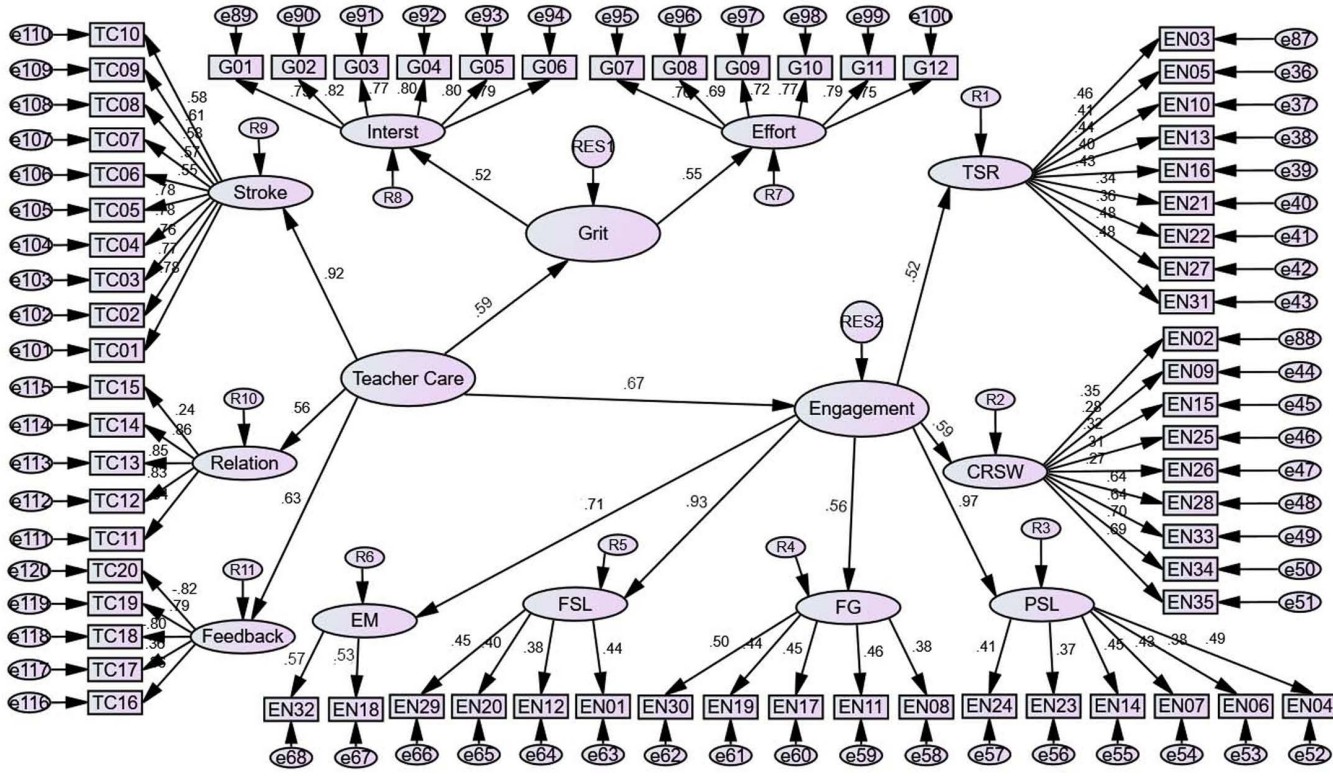

**Fig 1. The Final Measurement Model with Standardized Estimates.**

**Table 1. Assessment of the CFA Goodness of Fit.**

| Criteria | | Terrible | Threshold Satisfactory | Excellent | Evaluation |
|---|---|---|---|---|---|
| CMIN | 5720.793 | | | | |
| DF | 2130 | | | | |
| CMIN/DF | 2.686 | > 5 | > 3 | > 1 | Excellent |
| RMSEA | .065 | > 0.08 | < 0.08 | < 0.06 | Satisfactory |
| GFI | .932 | < 0.9 | > 0.9 | > 0.95 | Satisfactory |
| CFI | .930 | < 0.9 | > 0.9 | > 0.95 | Satisfactory |
| PNFI | .609 | < 0.5 | > 0.5 | | Excellent |
| TLI | .920 | > 0.9 | > 0.9 | > 0.95 | Satisfactory |

**Table 2. Composite Reliability and Discriminant Validity of the Factors.**

| | CR | AVE | MSV | MaxR(H) | Grit | Student Engagement | Teacher Care |
|---|---|---|---|---|---|---|---|
| Grit | 0.721 | 0.682 | 0.654 | 0.780 | 0.826 | | |
| Student Engagement | 0.850 | 0.772 | 0.703 | 0.940 | 0.847 | 0.879 | |
| Teacher Care | 0.727 | 0.913 | 0.774 | 0.861 | 1.002 | 0.827 | 0.956 |

Based on the results presented in Table 3, it is evident that teacher care is a significant predictor of both grit (t = 9.07, p = .000) and student engagement (t = 7.976, p = .000).

Table 4 presents the results of a structural model assessment, focusing on the relationships between two parameters, student engagement and grit. The structural model assessment highlights significant positive contributions of both student engagement (Estimate = .453, CI [.159,.356], $p$ = .001) and grit (Estimate = .354, CI [.256,.578], $p$ = .001) to the model. These results underscore the critical roles of engagement and grit in the underlying structure of the observed phenomena. The results reveal that teacher care plays a significant role in influencing students' engagement and grit, highlighting its importance in the educational context. Specifically, approximately 45% of the variations observed in students' engagement can be directly attributed to the effect of teacher care. This indicates that nearly half of the students' motivation, participation, and emotional investment in learning activities are shaped by how much they perceive their teachers as supportive and empathetic. Similarly, about 35% of the variations in students' grit, which encompasses perseverance and passion for long-term goals, are also linked to the influence of teacher care. This suggests that a caring teacher-student relationship not only fosters immediate academic engagement but also cultivates essential character traits that drive sustained effort and resilience. These findings emphasize the transformative potential of teacher care in enhancing both the emotional and behavioral dimensions of students' academic experiences. Moreover, the results align with existing research, which underscores the critical role of supportive teaching practices in fostering positive student outcomes, thereby offering valuable insights for educators and policymakers aiming to optimize learning environments.

## 5. Discussion

This research explored the power of teacher care in enhancing EFL learners' engagement and grit in China. The findings from the SEM analysis demonstrated significant paths through which teacher care influences EFL learners' engagement and grit. The SEM results indicated that teacher care directly predicts both engagement and grit among EFL learners, with about 59 percent of changes in grit and about 67 percent of changes in engagement being explained by teacher care. This robust correlation underscores the critical role of teacher care in fostering a supportive learning environment that enhances both emotional and behavioral student outcomes.

Regarding the role of teacher care on student engagement, the path analysis revealed that teacher care significantly enhances student engagement. When students perceive their teachers as caring, it provides them with emotional and instrumental support, which buffers against stress and promotes their involvement in academic activities. This finding is also justified in light of social support theory which suggests that when students perceive their teachers as caring and supportive, they have access to a valuable source of emotional and instrumental support. This support buffers the negative impact of stressors and ultimately contributes to their engagement [29].

**Table 3. The Results of Path Analysis with SEM.**

|  |  |  | Estimate | S.E. | C.R. | P |
|---|---|---|---|---|---|---|
| Grit | <— | Teacher Care | .593 | .040 | 9.079 | .000 |
| Student Engagement | <— | Teacher Care | .674 | .031 | 7.976 | .000 |

**Table 4. Structural Model Assessment.**

| PARAMETER | ESTIMATE | LOWER | UPPER | P |
|---|---|---|---|---|
| Student engagement | .453 | .159 | .356 | .001 |
| Grit | .354 | .256 | .578 | .001 |

Teacher care acts as a catalyst, enhancing students' emotional well-being and providing them with the psychological resources needed to engage actively in their learning and persist through difficulties. This interconnected influence is supported by the rhetorical/relational objective theory, which posits that teachers' communicative behaviors fulfill both educational objectives and students' psychosocial needs [56]. The impact of teacher care on the engagement of learners in striving for educational objectives in an EFL setting can be theoretically elucidated by considering the advancement of PP in language acquisition, employing a constructive approach for the growth of EFL educators and learners' biological, individual, and interpersonal aspects [9]. Teacher care is essential in PP, which emphasizes the ways individuals can flourish and experience greater happiness. This approach highlights positive emotions such as optimism, passion, adaptability, positivity, and similar feelings, rather than destructive emotions [3,58,65].

The findings maintain the rhetorical/relational objective theory and align with prior research that indicated elevated levels of teacher care resulted in enhanced performance, engagement, contentment, knowledge, well-being, and motivation among learners [28,60]. Therefore, it appears that when educators demonstrate caring actions such as displaying empathy towards learners, sustaining visual contact with everyone, and exhibiting genuine curiosity in their academic progress, learners' emotional and psychosocial requirements are fulfilled, making them more prone to attend classes consistently [23,50]. Based on this theory, the majority of educators' communication actions have the potential to fulfill the objectives of educators and cater to the requirements of students [58]. Consequently, it is contended that teacher care has been linked with several favorable results, such as enhanced attendance, heightened study duration, enhanced scholastic performance, reduced anxiety, and decreased attrition rate [66].

In line with SDT, teacher care stimulates learners' intrinsic motivation and increases their focus during the educational process, thereby resulting in increased engagement at a more advanced level. Based on SDT, educators are accountable for equipping learners with chances to embrace their principles and requirements and for offering captivating and contextually rich input and resources to learners [67]. As SDT focuses on how classroom settings affect individuals' emotional requirements, teacher care aids in fostering the self-autonomy of learners and motivates them to investigate on their own, enabling them to actively confront learning obstacles [68]. In the same vein, the significance of constructive teacher relational factors for improving students' achievement of positive educational results is demonstrated in the Xie and Derakhshan [69] statement that educators can establish a pleasant learning atmosphere by demonstrating regard and care for learners, fostering constructive teacher-student relations, and cultivating curiosity in the target language. These actions will finally assist students in maintaining their engagement in L2 learning. This outcome advocates the notion of Derakhshan et al. [60] who posited that learners' engagement in language courses could depend on the optimal connection between learner and instructor and the presence of considerable teacher care. Given that teacher affections have a significant impact on forming interpersonal connections with students in the classroom setting, their supportive actions can create an environment that promotes a positive bond with learners in the EFL class.

Regarding the role of teacher care on student grit, the SEM analysis also showed a significant path from teacher care to student grit. Students who feel cared for by their teachers are more likely to persevere through challenges and maintain their motivation over time. This relationship can be theoretically explained by PP and SDT. PP highlights the importance of positive emotions and supportive relationships in fostering resilience and perseverance [9]. SDT emphasizes the role of supportive environments in promoting intrinsic motivation and self-determination, leading to higher levels of grit [68].

In line with SDT, teacher care addresses three psychological needs, thereby increasing their level of grit. For instance, competence is cultivated when teachers provide positive feedback and set stimulating goals and when students believe they can deal with setbacks, they are more prone to persevere in their efforts, a key aspect of grit [68]. Indeed, the feedback and guidance provided by caring teachers help students develop resilience and a long-term commitment to their goals.Moreover, teacher care can inspire self-directed learning as it raises a sense of ownership and accountability towards one's learning, which drives long-standing commitment and perseverance [67].

Also, through the supportive relationships, students feel understood, valued, and connected to their teachers, they preserve their effort and interest over prolonged periods, even in the case of challenges [69]. Indeed, teachers' favorable interpersonal connections with the students can establish a trustworthy classroom environment that can ultimately enhance the probability of successful teaching and learning. In this kind of situation, students' persistence and enthusiasm for language learning are more likely to be cultivated. The favorable impact of teacher care on Chinese EFL learners' grit can be rationalized by the observation that learners who receive care and appreciation from their educators are more likely to persist in their learning endeavors despite encountering difficulties and obstacles [32]. The upshots are also in line with Eskreis-Winkler et al. [70] who specified that support from educators was meaningfully related to grit. Moreover, when learners perceived increased teacher assistance, the connection between grit and academic success was more robust [33]. The results of this study serve as an indirect confirmation of Pan's [4] research, which established a strong correlation between teachers' positive behaviors and students' levels of grit. It is pertinent to contextualize this finding in comparison to the results reported by Shen and Guo [61]. The present finding also validates the results reported by Feng et al. [71] as they established a significant correlation between the level of parental and teacher support and students' persistence and effort.

## 6. Conclusion

It is presumed that when there is good interaction between learners and educators, students can undergo positive emotions which are essential to successful teaching and learning [72]. Teacher care as a positive instructor communication can be elucidated in the context of PP which has garnered significant attention over the past two decades, encompassing three primary pillars; namely constructive encounters, constructive individual attributes, and constructive organizations [35]. Consistent with the results of a prior investigation conducted by Lavy and Naama-Ghanayim [32], the outcomes of the current study proved that there is a simultaneous occurrence of elevated perceptions of care and engagement. This can be academically illuminated by the interconnectedness of language education, which compels language teachers to utilize affirmative (non)verbal signals of communication and convey positive feelings and emotions to students within the language learning setting, thereby establishing a socially and interpersonally conducive learning context [22].

Likewise, as suggested by the advocates of PP in EFL education [34], since positive feelings of educators and relational actions are powerful predictors of any positive EFL learning concepts, they also foster learners' engagement [36]. When learners are instructed by teachers who are concerned about them, they experience positive educational results such as increased resourcefulness, greater contentment, and perseverance. These positive responses and enthusiasm are essential for their active engagement in the learning process and contribute to their grit. This is because when teachers identify and nurture students' aspirations, inclinations, and interests, they create supportive classroom environments that promote learning [73].

Teacher trainers are accountable for equipping L2 instructors who, among other duties, possess strong interpersonal skills in establishing a welcoming and friendly classroom environment, engaging in effective communication with learners, and demonstrating culturally sensitive care towards them [22]. Managers of language schools and committees responsible for recruiting language teachers can expand the range of criteria for selecting effective teachers in a manner that includes not only linguistic expertise, but also a strong comprehension of psychological, interpersonal, emotional, and cultural elements of language instruction. These elements are significant in defining the efficacy of the knowledge that EFL teachers impart to their learners [74].

The present study is subject to certain limitations, and it is advisable to exercise caution when generalizing the findings. The inclusion of self-reported data in testing the model necessitates a mixed-methods approach to obtain more comprehensive insights, ultimately enhancing our understanding of the relationships between key variables. Furthermore, it is important to note that all participants in this study were EFL students. This raises the need for further research to be conducted specifically with ESL teachers to validate the proposed model within this particular context. This study had

some other constraints that must be considered when describing the results obtained. For example, this study employed a quantitative approach; thus, additional inquiries utilizing a combination of methods are required to provide a more holistic view of the association between teacher care, grit, and engagement.

In addition, this research did not investigate the demographic factors of college educators or students and their possible correlations with the study's variables; hence, these factors could present an interesting line for future investigation. It would be noteworthy to assess the moderating effects of issues such as age, level of education, and area of specialization on the relationships between the concepts. Additional research will investigate the enabling or regulating observable actions of more resilient students that will yield academic advancements of resilience and contribute to the development of grit-improving treatment programs. It is advisable to conduct a longitudinal form of investigation because, firstly, time plays a crucial role in acquiring more evident and advantageous information from the subjects [75], and secondly, the grit scale intends to assess an individual's ambition and persistence towards enduring goals.

## Author contributions

**Conceptualization:** Jianguo Xiao.

**Data curation:** Xin Wang, Jianguo Xiao.

**Funding acquisition:** Jianguo Xiao.

**Investigation:** Xin Wang.

**Methodology:** Xin Wang, Jianguo Xiao.

**Writing – original draft:** Xin Wang, Jianguo Xiao.

**Writing – review & editing:** Xin Wang, Jianguo Xiao.

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
