## [Decision Letter · Decision Letter 0]

Dear Dr. Xiao,

Thank you for submitting your manuscript to PLOS ONE. After careful consideration, we feel that it has merit but does not fully meet PLOS ONE’s publication criteria as it currently stands. Therefore, we invite you to submit a revised version of the manuscript that addresses the points raised during the review process.

We look forward to receiving your revised manuscript.

Kind regards,

Zhanni Luo

Academic Editor

PLOS ONE

Journal Requirements:

https://doi.org/10.3389/fpsyg.2021.601992

In your revision ensure you cite all your sources (including your own works), and quote or rephrase any duplicated text outside the methods section. Further consideration is dependent on these concerns being addressed.

The research is supported by: the General Program Supported by The National Social Science Fund of China in 2020, Research on Improving the Long-term Mechanism of Training Socialist Builders and Successors, (No.20BKS140).  

6. Please ensure that you refer to Figure 1 and 2 in your text as, if accepted, production will need this reference to link the reader to the figure.

Additional Editor Comments :

See reviewer comments.

Reviewers' comments:

Reviewer's Responses to Questions

**Comments to the Author**

1. Is the manuscript technically sound, and do the data support the conclusions?

Reviewer #1: Partly

Reviewer #2: Partly

Reviewer #3: Partly

Reviewer #4: Partly

2. Has the statistical analysis been performed appropriately and rigorously?

Reviewer #1: No

Reviewer #2: No

Reviewer #3: No

Reviewer #4: Yes

3. Have the authors made all data underlying the findings in their manuscript fully available?

Reviewer #1: No

Reviewer #2: Yes

Reviewer #3: Yes

Reviewer #4: No

4. Is the manuscript presented in an intelligible fashion and written in standard English?

Reviewer #1: No

Reviewer #2: No

Reviewer #3: No

Reviewer #4: No

Reviewer #1: Comments on Method:

1. The term "convenient sampling" should be revised to "convenience sampling"

2. The sample size is different between the "401" shown in the database and the "264" mentioned in the study

3. The three scales used in the study have not been adapted to fit the specific EFL learning context

4. The Procedure section lacks sufficient detail, leaving gaps in the research process description

5. There is inconsistency in the Instruments section, with terms such as instrument, scale, and questionnaire used interchangeably.

6. Despite its mention in the Literature Review, social engagement was not measured in the "Student Engagement" scale. However, there is no explanation provided for this omission or removal.

Comments on Results and Discussion:

7. The reliability of overall questionnaire is missing

8. The MIN/DF evaluation result reported in Table 1 is not in line with the standard

9. The factor loadings are not presented when testing for convergent validity

10. The statement "the model has attained CR" is inaccurate, and actually, it should be rephrased to “the model meets the standard value for CR”

11. There is no description about the result of discriminant validity

12. The AVE square root for Student Engagement in Table 2 may be miscalculated, and the correct value is √0.717 = 0.847, not 0.879

13. Table 3 reflects the results of "Path Analysis" rather than "Linear Regression," as it simultaneously analyzes the direct impact of one independent variable (Teacher Care) on two dependent variables (Grit and Student Engagement).

14. The same construct "Student Engagement" is written in two ways in Table 3: "Learner Engagement" and "Student Engagement"

15. The Discussion section lacks focus and organization, with a scattered presentation of findings. Thus, classification based on the theme will make it more understandable and logical.

Other: The word "prosperous” in “prosperous language instruction" can be replaced with "effective" or "successful."

Reviewer #2: In the manuscript titled The Relationship between Students’ Engagement, Grit, and Teacher Care in Chinese Context, Xin Wang and Jianguo Xiao explore the interplay between teacher care, grit, and engagement among Chinese EFL (English as a Foreign Language) learners using Positive Psychology (PP) and Structural Equation Modeling (SEM). The results indicate a strong predictive relationship between teacher care and both engagement (67%) and grit (59%). The study effectively integrates PP theory with self-determination theory (SDT) to validate the psychological impact of teacher care in educational settings. However, the exclusive use of quantitative methods omits rich qualitative insights that could have contextualized the findings more deeply, it narrows the interpretative scope of the relationship dynamics between teacher care, grit, and engagement. Therefore, a minor revision has to be done before this manuscript can be accepted for publication in PLOS ONE.

Major Comments:

1. The sample consists solely of Chinese EFL learners, leaving unverified applicability to other cultural or academic contexts.

2. Factors such as age, educational level, or specialization were not considered, which might influence the predictive roles of teacher care.

3. The current manuscript needs to be polished by a native English speaker or a professional language editing service.

4. Relying on self-reported measures introduces potential biases (e.g., social desirability, and subjective interpretation).

5. The study's cross-sectional nature limits the ability to establish causality, though the SEM analysis supports robust correlations.

6. The literature review section of the paper does not clearly summarize the proposed hypotheses. As a result, the discussion section lacks structure and logical coherence, making it appear disorganized.

Minor Comments:

1. The abstract is generally well-written but includes minor grammatical inaccuracies. In line 3, "Also, the focus of learning is being given to the key function that grit has in the path of language education as it affects their success all through the path of education."

2. Maintain consistency in terminology. For example, "grit" is sometimes called "perseverance," which might confuse readers. Use a single term unless intentionally distinguishing concepts.

3. Ensure uniform use of past tense when describing methods or completed results (e.g., "were examined" instead of "are examined").

4. Some references (e.g., Pishghadam et al., 2019) lack detailed citation formatting or consistency in DOI inclusion. Ensure all references adhere strictly to the journal’s referencing guidelines (e.g., APA, MLA, etc.). For instance: "Dincer et al. (2019)" in the main text should correspond precisely to the format and details in the reference list.

5. The resolution of the included figures (e.g., structural equation modeling diagrams) is suboptimal. Ensure high-resolution graphics for clarity, particularly for SEM paths. Ensure consistent formatting in figure titles and legends. For example, "Figure 1. The Final Measurement Model with Standardized Estimates" can be improved to explain the figure’s purpose briefly.

6. Ensure the keywords are in alphabetical order and consistent in capitalization (e.g., "Positive Psychology" instead of "positive psychology").

7. Use consistent formatting for all headings and subheadings (e.g., font size, bold/italic styles).

8. In Figure 3, the direction of the arrows needs to be reversed. Additionally, it would be helpful to label the hypothetical pathways clearly and indicate within the table whether each hypothesis was supported or not.

Reviewer #3: Research on the relationship between student engagement, grit and teacher care is of importance in the field of EFL education and the manuscript is of value. Yet, there are some suggestions.

1.It is suggested that the authors rewrite the abstract. The research background, methodology, and major findings should be clearly stated in abstract.

2.In the manuscript, there are different expressions like “teacher care”, “teacher caring”, “educator care”. Consistency is highly recommended for key concepts.

3.Though many references are cited in the introduction, the research background, purpose and significance are not concretely and clearly stated. The authors are advised to write a more proper introduction.

4.The subtitle of Part 2 should be “Literature Review”.

5.In 2.1, 2.2 and 2.3, the authors mainly define the key concepts of student engagement, grit and teacher care. The literature review should be research question oriented and it is suggested that more relevant researches on the relationship between teacher care, student engagement, and grit should be reviewed.

6.On Page 5, the authors write “Educators are widely recognized as influential contributors…while demonstrating a genuine concern for these interactions”. How is this related to grit?

7.In Line 7, Page 7, there is a sentence “Motivated by the gap…”. However, it seems no research gap can be clearly identified from the reviewed literature. The authors are advised to supplement some existing literature concerned the relationship between teacher care, student engagement, and grit, from which RQs can be naturally derived.

8.The manuscript is seemed to explore the relationship between teacher care, student engagement and grit. Is there any relationship? It is suggested that theoretical basis and the framework of transmission mechanism before SEM should be added.

9.In Part 3, the authors fail to describe the methodology properly. 1) Participants are not clearly described. What does “eleven regions” refer to? Why these regions? What does the author mean by “university or school level”? 2) Instruments are not properly introduced. There are some contradictory expressions such as in 3.2.1, the authors write “it aims to assess three distinct aspects of emotional engagement”. Yet, I can only see two aspects “the bond between educators and learners” and “the support received from peers and guardians”. 3) In the last sentence in 3.3, there is “The pupils…”, while the authors have explained in 3.1 the participants are aged from 18 to 33. 4) I cannot catch what the authors are trying to do in 3.4 Statistical Analysis. More details should be provided. 5) Descriptive analysis of the data should be supplemented in Part 3.

10.It seems that Part 4 is merely composed of three tables and two figures. It is suggested more result explanations should be supplemented.

11.Discussion should be based on the results of SEM. The influencing paths and mechanism should be explicitly discussed in Part 5.

12.On page 10-11, “It is evident that when educators demonstrate care …performance and cognitive capability can be readily enhanced”. However, performance and cognitive capability are not mentioned at all in the model. Meanwhile, it is suggested that all the variables in the model should be explicitly listed and explained.

13.It is suggested that subtitles be added in Part 5 and 6 to enhance readability.

14.Language throughout the manuscript should be extensively revised so as to meet the requirements of an academic journal.

Reviewer #4: 1. The manuscript lacks a section dedicated to hypothesis development, where the authors should propose hypotheses based on a comprehensive literature review.

2. You suggest that student engagement consists of six elements: future goals, control of schoolwork, extrinsic motivation, family support, peer support, and teacher-student interactions. Why? It is necessary to discuss the rationale behind this composition, ideally grounded in one or more theoretical models.

3. Does this framework genuinely based on the self-determined theory (SDT)? The self-determined theory (SDT) encompasses three elements: competence, autonomy, and relatedness. Are these three elements incorporated into your framework?

4. If extrinsic motivation can enhance student engagement, what about intrinsic motivation? Why is intrinsic motivation omitted from the framework?

5. The self-determined theory (SDT) was not introduced by Ryan and Deci in 2020; it was proposed approximately four decades ago. Although these authors did publish a related article in 2020, it is imperative for academic manuscripts to cite the foundational literature when referencing significant theories.

6. There is a logical gap in the manuscript. In section 2.3, you reference the self-determined theory (SDT) without explaining what it is or how it relates to teacher care (the focus of that section). While you mention "support from others," there is a logical gap between teacher support and teacher care that the authors need to address.

7. Section 2.4 is not justified. A literature review typically involves examining related studies, making the inclusion of a separate subsection for "related studies" unnecessary and odd.

8. There is an issue with the formulation of the first research question. Please reword the first research question.

9. The data analysis subsection within the "Methods" section is inadequate. Please provide more specificity in the data analysis process. The criterion is whether readers, without looking at the findings section, can understand the steps of your data analysis from the methods section alone.

10. It is recommended that the Conclusion section be organized in the "summary-implication-limitations-future work" structure.

11. The Introduction section should follow the structure of "background-gaps-aims and significance-research questions." In the current manuscript, the research gaps are not clearly highlighted.

12. The final paragraph of the Introduction is excessively long; it is advised to break it into smaller sections.

13. Sections 2.1 and 2.2 each contain a single paragraph that is excessively long. It is recommended that these paragraphs be broken down into shorter, more manageable sections.

14. Please provide the full name when any abbreviation is used for the first time, such as SDT.

15. The capitalization of keywords is inconsistent.

**Do you want your identity to be public for this peer review?** For information about this choice, including consent withdrawal, please see our Privacy Policy

Reviewer #1: No

Reviewer #2: No

Reviewer #3: No

Reviewer #4: No

---

## [Author Response · Author response to Decision Letter 1]

18 Feb 2025

Please see the file "Response to Reviewers"

---

## [Decision Letter · Decision Letter 1]

Dear Dr. Xiao,

Thank you for submitting your manuscript to PLOS ONE. After careful consideration, we feel that it has merit but does not fully meet PLOS ONE’s publication criteria as it currently stands. Therefore, we invite you to submit a revised version of the manuscript that addresses the points raised during the review process.

We look forward to receiving your revised manuscript.

Kind regards,

Zhanni Luo

Academic Editor

PLOS ONE

Journal Requirements:

**Additional Editor Comments:**

Thank reviewers for their valuable feedback. Additionally, please pay attention to the following points:

1. Table Sequence: In the manuscript, Table 3 should be followed by Table 4, but it is currently followed by Table 1. Please revise.

2. Table Headers: Each table should include clear and descriptive headers. For example, a header such as "gender" clarifies that the entries should be "male" or "female." Currently, the second column of Table 1 and the first column of Table 2 lack headers. Please ensure all columns are appropriately labeled.

3. RMSEA Benchmark: In Table 1, the RMSEA benchmark is cited as 0.06. However, based on my review of the literature, the accepted benchmark is 0.05. Please either provide a credible reference supporting the 0.06 threshold or revise the value to 0.05 after verification.

4. Model Fit Interpretation: The interpretation of model fit indices requires revision. For instance, a CMIN/df value between 1 and 3 is typically considered excellent, rather than the current interpretation provided. Please carefully review and update the criteria used for evaluating model fit.

5. Instrument presentation. While the sources of each scale are clearly stated, enhancing transparency and reproducibility would be beneficial. Consider adding a table listing all survey items used in the study. If this table is included, please ensure each item is appropriate and accurately represents the constructs measured. This is a suggestion for improvement (not mandatory).

6. Long paragraphs. Some paragraphs in the introduction are excessively long, spanning half a page or more. To improve readability and logical flow, consider breaking these into shorter, more focused paragraphs. This will help emphasize key points and enhance the overall coherence of the section. This is a suggestion for improvement (not mandatory).

7. Visual presentation: The current presentation of Figures 1 and 2 does not meet standard aesthetic guidelines. For consistency and clarity:

- Ensure graphical elements representing concepts at the same level are of uniform size.

- Align items vertically unless there is a specific rationale for deviation.

- Distribute lines and elements evenly to avoid visual imbalance.

For example, in Figure 1, the element "student engagement" connects to "future goals" and "teacher-student interactions," but the boxes for these elements are inconsistently sized, misaligned, and unevenly spaced relative to the central element. Please revise these figures to adhere to best practices in visual presentation.

Reviewers' comments:

Reviewer's Responses to Questions

**Comments to the Author**

Reviewer #1: All comments have been addressed

Reviewer #3: All comments have been addressed

2. Is the manuscript technically sound, and do the data support the conclusions?

Reviewer #1: Yes

Reviewer #3: Partly

3. Has the statistical analysis been performed appropriately and rigorously?

Reviewer #1: Yes

Reviewer #3: No

4. Have the authors made all data underlying the findings in their manuscript fully available?

Reviewer #1: Yes

Reviewer #3: Yes

5. Is the manuscript presented in an intelligible fashion and written in standard English?

Reviewer #1: Yes

Reviewer #3: Yes

Reviewer #1: The revised manuscript shows significant improvements, particularly in the methodology and discussion sections, and the major concerns have been appropriately resolved. However, I recommend a thorough proofreading to enhance linguistic and logic accuracy. For example:

1. The title of Table 3 contains a punctuation mistake: The Results of "Path Analysis with SEM.

2. In the discussion of CMIN/DF (2.686), it should be described as "excellent" rather than "satisfactory", in accordance with the standard model fit criteria you present in Table 1. Because the standard in your Table shows that ">3 is satisfactory, >1 is excellent".

Reviewer #3: 1. The whole section of “Literature Review” should be the review of related studies. In this case, the subtitle of 2.4 “Related Studies” is improper.

2. In this study, two research questions and two research hypotheses were proposed. From RQ1, it appears that the authors aim to explore the relationship among “Chinese EFL students’ engagement, grit, and teacher care”, while H01 states that there is no relationship among them. This is quite odd and perplexing. The premise of RQ2 is that teacher care can predict student engagement and grit among Chinese EFL learners, but it is hypothesized in H02 “Teacher care does not significantly predict student engagement and grit among Chinese EFL learners”.

Moreover, in the results and discussion sections, the authors primarily focus on addressing the research questions, with nothing concerning the research hypotheses. In fact, as the authors stated the research indeed affirms that “teacher care plays a significant role in influencing students' engagement and grit”, which rebuts H01. Therefore, it is strongly recommended that the authors re-consider the logical consistency between RQs and the necessity of proposing the hypotheses.

3. Please modify the table number. There are two tables numbered “Table 1”, the second of which should be “Table 4”.

4. There exist inaccuracy and inconsistency in some sentences like “The results of Table 3 signify that about 59 percent of variations in the EFL students’ grit can be predicted by their teacher care, and about 67 percent of variations in the EFL student engagement can be predicted by their teachers’ care.” Please check the language to ensure consistency and accuracy.

**Do you want your identity to be public for this peer review?** For information about this choice, including consent withdrawal, please see our Privacy Policy

Reviewer #1: No

Reviewer #3: No

---

## [Editor Report · Decision Letter 2]

The Relationship between Students’ Engagement, Grit, and Teacher Care in Chinese Context

PONE-D-24-49069R2

Dear Dr. Xiao,

We’re pleased to inform you that your manuscript has been judged scientifically suitable for publication and will be formally accepted for publication once it meets all outstanding technical requirements.

Kind regards,

Zhanni Luo

Academic Editor

PLOS ONE

Additional Editor Comments (optional):

Thank you for your thorough revisions to the manuscript. We acknowledge the significant efforts you have made to address all of the reviewers' comments in detail. Your responsiveness has greatly strengthened the paper.
---

## [Editor Report · Acceptance letter]

PONE-D-24-49069R2

PLOS ONE

Dear Dr. Xiao,

I'm pleased to inform you that your manuscript has been deemed suitable for publication in PLOS ONE. Congratulations! Your manuscript is now being handed over to our production team.

Kind regards,

on behalf of

Dr. Zhanni Luo

Academic Editor

PLOS ONE